# Two-dimensional shear wave elastography and ultrasound-guided attenuation parameter for progressive non-alcoholic steatohepatitis

Hidekatsu Kuroda[1]*, Yudai Fujiwara[1], Tamami Abe[1], Tomoaki Nagasawa[1], Takuma Oguri[2], Sachiyo Noguchi[2], Naohisa Kamiyama[2], Yasuhiro Takikawa[1]

1 Division of Hepatology, Department of Internal Medicine, Iwate Medical University, Iwate, Japan,
2 Ultrasound General Imaging, GE Healthcare, Hino, Tokyo, Japan

* hikuro@iwate-med.ac.jp

**Data Availability Statement:** All relevant data are within the manuscript and its Supporting information files.

## Abstract

### Background and aims

We investigated the usefulness of combining two-dimensional shear wave elastography and the ultrasound-guided attenuation parameter for assessing the risk of progressive non-alcoholic steatohepatitis, defined as non-alcoholic steatohepatitis with a non-alcoholic fatty liver disease activity score of $\geq$4 and a fibrosis stage of $\geq$2.

### Methods

This prospective study included 202 patients with non-alcoholic fatty liver disease who underwent two-dimensional shear wave elastography, ultrasound-guided attenuation parameter, vibration-controlled transient elastography, the controlled attenuation parameter, and liver biopsy on the same day. Patients were grouped according to liver stiffness measurement using two-dimensional shear wave elastography and the attenuation coefficient, assessed using the ultrasound-guided attenuation parameter: A, low liver stiffness measurement/low attenuation coefficient; B, low liver stiffness measurement/high attenuation coefficient; C, high liver stiffness measurement/low attenuation coefficient; and D, high liver stiffness measurement/high attenuation coefficient.

### Results

Two-dimensional shear wave elastography and vibration-controlled transient elastography had equivalent diagnostic performance for fibrosis. The areas under the curve of the ultrasound-guided attenuation parameter for identifying steatosis grades $\geq$S1, $\geq$S2, and S3 were 0.89, 0.91, and 0.92, respectively, which were significantly better than those of the controlled attenuation parameter (P<0.05). The percentages of progressive non-alcoholic steatohepatitis in Groups A, B, C, and D were 0.0%, 7.7%, 35.7%, and 50.0%, respectively (P<0.001). The prediction model was established as logit (p) = 0.5414 × liver stiffness measurement (kPa) + 7.791 × attenuation coefficient (dB/cm/MHz)—8.401, with area under the

**Funding:** This work was supported by JSPS KAKENHI [grant number JP 19K08400]. T.O., S.N. and N.K. are paid employees of GE Healthcare Japan. The funder provided support in the form of salaries for authors [H.K.]. However, it did not have any additional role in the study design, data collection, and analysis, decision to publish, or preparation of the manuscript. The specific roles of these authors are articulated in the 'author contributions' section.

**Competing interests:** The authors have read the journal's policy, and the authors of this manuscript have the following competing interests: T.O., S.N. and N.K. are paid employees of GE Healthcare Japan. This does not alter our adherence to PLOS ONE policies on sharing data and materials. The other authors declare that they have no competing interests. There are no products in development or marketing products to declare.

receiver operating characteristic curve, sensitivity, and specificity values of 0.832, 80.9%, and 74.6%, respectively; there was no significant difference from the FibroScan-aspartate aminotransferase score.

## Conclusion

Combined assessment by two-dimensional shear wave elastography and the ultrasound-guided attenuation parameter is useful for risk stratification of progressive non-alcoholic steatohepatitis and may be convenient for evaluating the necessity of specialist referral and liver biopsy.

## Introduction

Non-alcoholic fatty liver disease (NAFLD) is rapidly becoming a global public health problem [1,2]. NAFLD is categorized as either non-alcoholic fatty liver (hepatis steatosis without hepatocellular injury) or non-alcoholic steatohepatitis (NASH) (hepatic steatosis and inflammation with hepatocyte injury, with or without fibrosis) [3–5]. Because NASH leads to fibrosis, cirrhosis, and hepatocellular carcinoma, developing a clinical index capable of efficiently extracting NASH among NAFLD patients, who might benefit from new pharmacotherapy treatments and determination of the presence or degree of inflammatory liver injury, is desired [6–10].

Liver biopsy (LB), the gold standard for diagnosing NASH [11], has disadvantages, including risk of bleeding, sampling error, and inter-pathologist variability [12]. Recent innovations in imaging have demonstrated their potential in diagnosing NASH and advanced fibrosis [13]. Newsome et al. reported that the FibroScan-aspartate aminotransferase (FAST) score noninvasively and efficiently identifies patients at risk for advanced NASH [14]. The FAST score combines two physical biomarkers, liver stiffness measurement (LSM) (assessed using vibration-controlled transient elastography [VCTE]) and hepatic steatosis (assessed using the controlled attenuation parameter [CAP]), with aspartate aminotransferase levels, into a single score [14,15]. This study concluded that this score reduces unnecessary LB in patients unlikely to have significant disease [14].

Recent investigations have demonstrated that ultrasound-based two-dimensional shear wave elastography (2D SWE) is a promising alternative to VCTE for evaluating liver fibrosis. Several studies reported that LSM obtained by 2D SWE, VCTE, and magnetic resonance elastography performs well in diagnosing advanced fibrosis in NAFLD patients [16–19]. Other studies that used LB as the reference standard reported good performance of CAP for grading liver steatosis [20–22]. However, CAP performance was poor in patients with higher body mass indices (BMI) due to subcutaneous adipose tissue thickness [20,22,23]. Moreover, as CAP can only be performed in A-mode, its main limitation is insufficient visual guidance. Previously, we reported the development of the ultrasound-guided attenuation parameter (UGAP), which measures the attenuation coefficient (AC) (dB/cm/MHz) of the B-mode ultrasonic signal via general ultrasonography [24]. We showed that UGAP is more useful than CAP in estimating hepatic steatosis severity in chronic liver disease patients. To the best of our knowledge, no previous prospective cohort study has investigated the usefulness of a composite approach using ultrasound parameters for assessing the progressive NASH risk in biopsy-confirmed NAFLD patients.

Therefore, we hypothesized that a combined evaluation by 2D SWE and UGAP would improve quality of care for NAFLD patients and could be useful in progressive NASH risk assessment.

## Materials and methods

### Patients

This cross-sectional, prospective study included patients evaluated at Iwate Medical University Hospital, Morioka, Japan. The cohort consisted of 233 consecutive NAFLD patients who underwent 2D SWE, UGAP, VCTE, CAP, and LB between April 2016 and March 2020. The inclusion criteria were ability to provide informed consent and age from 18 to 80 years. NAFLD diagnosis was based on steatosis presence on LB. The exclusion criteria were alcohol use (consuming ≥40 g alcohol/day for men and ≥20 g/day for women in the preceding 12 months) and other liver diseases, such as chronic hepatitis, drug use associated with fatty liver, or untreated hypothyroidism. Progressive NASH was defined as NASH in patients who also had an elevated NAFLD activity score (NAS ≥ 4) and advanced fibrosis (stage 2 or higher [F ≥ 2]) [14].

The control group comprised 20 participants with mean age and sex ratios matched to those of the non-alcoholic fatty liver and NASH groups; all participants had normal liver enzyme levels, with no evidence of a fatty liver.

The study was approved by the local Ethics Committee of Iwate Medical University (H26-124). Patients provided written informed consent, in accordance with the ethical standards laid down in the 1964 Declaration of Helsinki and its later amendments.

### Liver stiffness measurements and attenuation parameters

**2D SWE.** The 2D SWE, UGAP, VCTE, and CAP assessments were performed independently by one of two experienced radiologists in control participants and NAFLD patients on the same day as LB. Radiologists were blinded to patients' histological and clinical data. 2D SWE and UGAP were performed using the LOGIQ E9 XDclear 2.0 ultrasound scanner (GE Healthcare, Wauwatosa, WI, USA), with a C1-6-D convex array probe, at a frequency of 4.0 MHz. LSM was performed using 2D SWE [25,26]. Patients were in the supine position with their right upper extremity lifted under fasting conditions for 4 hours. The liver target area was placed under guidance of a conventional, real-time, B-mode image. Scanning was performed between ribs in the right liver lobe (segment 5), with minimal scanning pressure applied. The color-coded map, approximately 30 mm × 15 mm in area, was placed at least 10 mm below the liver surface in an area of liver parenchyma free of large vessels. The ultrasound scanning probe was positioned in the intercostal spaces of the right liver lobe. A circular region of interest (ROI) with a 10-mm diameter was drawn inside the stiffness sample box, and mean liver stiffness value in the ROI was displayed. The sample box was adjusted to avoid vessel inclusion (S1 Fig). One LSM was obtained from each 2D SWE image. The median value of 10 LSMs was used to represent liver stiffness. LSM was considered invalid if 2D elasticity color signals were obtained in <50% of the map. After recording all LSMs, data were screened based on the following criteria: ≥10 valid measurements, 60% success rate (ratio of valid acquisitions to total acquisitions), and interquartile range (IQR) of <30% of the median LSM.

**UGAP.** The technique and procedure of UGAP assessment have been previously described [24]. The attenuation coefficient (AC) was calculated based on the reference phantom method reported by Yao et al. [27]. This method utilizes an ultrasound phantom with known attenuation and backscatter coefficients to compensate for the characteristics of transmission. The UGAP procedure is detailed in S2 Fig. We acquired B-mode image data of the

liver parenchyma (Segment 5), the same area evaluated using 2D SWE, VCTE, CAP, and biopsy. Radiofrequency-based ultrasound echo signals were analyzed by a dedicated prototype software program in MATLAB (MathWorks, Inc., Natick, MS). One of three ultrasound engineers (T.O., S.N., and N.K.) opened each image and set the ROI on the liver parenchymal area, avoiding vessels and setting it at least 20 mm from the liver surface. The engineers were blinded to all patient data. The AC was calculated based on the signal's decay slope between 55 and 120 mm of depth. A reliable AC was defined as >5 valid shots, 60% success rate, and IQR <30% of the median AC value.

**VCTE and CAP.** VCTE and CAP measurements were performed using the FibroScan® 502 Touch with a 3.5-MHz M probe (Echosens, Paris, France) [25,26]. A vibration of medium amplitude and low frequency is transmitted from the vibrator to the tissue by the transducer itself; this induces the propagation of an elastic shear wave through the liver tissue. The speed of the propagating wave is estimated using a 1-dimensional ultrasound technique, and is automatically converted to a measurement in terms of Young's modulus in units of kilopascals (kPa). In accordance with the manufacturer's guidance, all patients fasted for at least 4 hours before examination and were in the supine position with their right arm in abduction during measurement. The tip of the ultrasound probe is placed on the skin in an intercostal space overlying the right lobe of the liver (segment 5). A time-motion ultrasound image allows the operator to locate a portion of liver at least 6 cm thick and free of large vascular structures or ribs. The median and IQR value of successful LSMs (target ≥10) is calculated by the machine and recorded. VCTE measurements with ≥10 valid shots and 60% success rate were considered reliable and used for statistical analysis. If the LSM was valid, as measured in the same volume of the liver parenchyma between 25 and 65 mm in depth, the corresponding median CAP value was also considered reliable.

**FAST score.** The FAST score was calculated for each patient based on LSM results, CAP value, and aspartate aminotransferase value [14,15]. Its predictive performance and the prediction model using 2D SWE and UGAP were assessed using receiver operating characteristic (ROC) analysis.

## Laboratory data and histopathologic evaluation

Laboratory data, including total bilirubin, aspartate aminotransferase, alanine aminotransferase, albumin, gamma-glutamyl transpeptidase, fasting plasma glucose, immunoreactive insulin, high-density lipoprotein cholesterol, triglycerides, and platelet count, were recorded on the LB procedure day under fasting conditions. Type II diabetes mellitus was diagnosed based on the American Diabetes Association criteria [28]. Hypertension was diagnosed when the average of two or more diastolic blood pressure measurements on at least two subsequent visits was ≥90 mm Hg or when the average of multiple systolic blood pressure readings on two or more subsequent visits was consistently ≥140 mm Hg [29]. Dyslipidemia was diagnosed at low-density lipoprotein >140 mg/dL, high-density lipoprotein <40 mg/dL, or triglyceride >150 mg/dL [30].

Echo-assisted LB samples were obtained from patients using a 14-gauge needle biopsy kit. An adequate LB sample was defined as being >15 mm in length and/or having >6 portal tracts under a microscope. LB specimens were fixed in formalin, embedded in paraffin, stained with hematoxylin/eosin and gomori trichrome for fibrosis evaluation, and assessed by two experienced pathologists blinded to each other's readings and to patients' clinical and 2D SWE and UGAP data. A third observer evaluated 50 cases with significant inter-observer differences. Macrovesicular steatosis affecting ≥5% of hepatocytes was observed in all NAFLD patients. NASH was diagnosed using the fatty liver inhibition of progression algorithm and the

steatosis, activity, and fibrosis score [31]. Steatosis (1–3), ballooning (0–2), lobular inflammation (0–3), fibrosis (0–4), and NAS were scored using the NASH Clinical Research Network Scoring System.

## Statistical analysis

Statistical analyses were performed using SPSS (version 23; IBM, Armonk, NY, USA) and XLSTAT 2019 (Microsoft®, WA, USA). In our study, predetermined sensitivity and specificity values for UGAP were 85.7% and 81.5%, respectively. Significance level was 0.05, and margin of error was set to ±5%, yielding results accurate to within ±5% points [24]. Based on the formula described by Karimollah et al [32], sample sizes for sensitivity and specificity were 176 and 201, respectively. Thus, a sample size of 201 was finally selected. Data are presented as means ± standard deviations (normally distributed data) or as medians [25-75th percentiles] (non-normally distributed data). The Mann-Whitney U test was used to compare differences between two groups. The Kruskal-Wallis test was used to test differences among more than two independent groups.

Logistic regression models were used to examine factors associated with progressive NASH. Variables included age, sex, BMI, laboratory data, LSM using 2D SWE, and AC. Variables exhibiting P-values ≤0.05 in the univariate model were incorporated into the final stepwise logistic regression analysis. Fitness of each logistic model was verified by the Hosmer–Lemeshow test. ROC curves were constructed. Area under the ROC curve (AUROC) was calculated using the trapezoidal rule.

Differences in diagnostic accuracy of the groups were investigated by comparing the area under the curve (AUC) [33]. Optimal cut-off points for predicting different steatosis grades were identified from the highest Youden index. Sensitivity, specificity, positive and negative predictive values (PPV and NPV, respectively), and positive and negative likelihood ratios (LR + and LR−, respectively) were calculated using cut-offs obtained by the ROC curves. P-values <0.05 were considered statistically significant.

## Results

### Patients' baseline characteristics

A total of 233 NAFLD patients who underwent 2D SWE, UGAP, VCTE, CAP, and LB assessments were enrolled. Nineteen were excluded due to disqualified biopsy specimens (n = 15), excessive alcohol consumption (n = 3), and presence of autoimmune hepatitis (n = 1). 2D SWE, UGAP, VCTE, and CAP success rates were 95.3% (204/214), 100% (214/214), 94.4% (202/214), and 94.4% (202/214), respectively. CAP failed in 12 patients (BMI >30 kg/m², 9 patients; inability to optimally perform a breath hold, 3 patients). Statistical analysis included 202 (94.4%) patients (Fig 1). Table 1 summarizes characteristics of the included patients and the 20 control participants.

### Liver fibrosis assessment in NAFLD patients using 2D SWE and VCTE

LSM was performed using 2D SWE and VCTE in NAFLD patients to assess liver fibrosis stage. Median LSMs obtained by 2D SWE for stages (control), F0, F1, F2, F3, and F4 were (4.61), 6.04, 6.81, 8.56, 10.42, and 14.59 kPa, respectively, demonstrating a stepwise increase with increasing fibrosis severity (P < 0.0001) (S1 Table). Median LSMs obtained by VCTE for these stages were (5.48), 7.86, 8.16, 9.58, 14.90, and 23.66 kPa, respectively, also demonstrating a stepwise increase with increasing fibrosis severity (P < 0.0001). To investigate performance accuracy of 2D SWE or VCTE for liver fibrosis diagnosis in NAFLD patients, we calculated

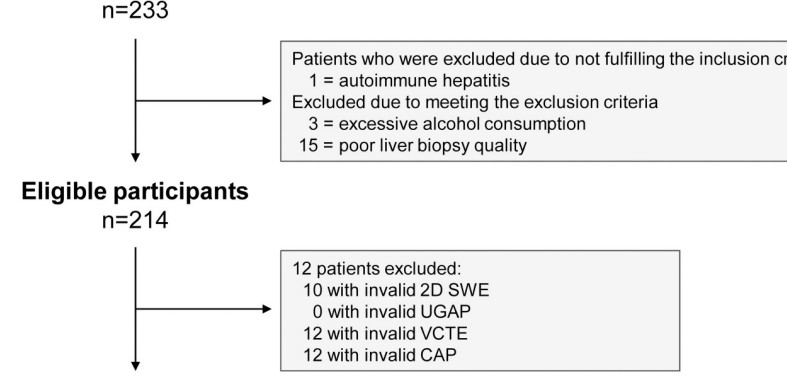

**NAFLD patients who underwent 2D SWE, UGAP, VCTE, CAP, and LB**
（April 2016 - March 2020）
n=233

Patients who were excluded due to not fulfilling the inclusion criteria
1 = autoimmune hepatitis
Excluded due to meeting the exclusion criteria
3 = excessive alcohol consumption
15 = poor liver biopsy quality

**Eligible participants**
n=214

12 patients excluded:
10 with invalid 2D SWE
0 with invalid UGAP
12 with invalid VCTE
12 with invalid CAP

**NAFLD patients available for analysis**
n=202

**Fig 1. Flow diagram of the study population.** 2D SWE, 2D shear wave elastography; UGAP, ultrasound-guided attenuation parameter; VCTE, vibration-controlled transient elastography; CAP, controlled attenuation parameter; LB, liver biopsy.

AUROC curve, potential cutoff, sensitivity, specificity, PPV, NPV, LR+, and LR− values for each liver fibrosis stage (Table 2). AUCs were higher for 2D SWE than for VCTE in all stages, but this was not statistically significant.

## Steatosis assessment in NAFLD patients using UGAP and CAP

AC was measured using UGAP. CAP-based measurements obtained using VCTE were compared with the steatosis grade. Median ACs for grade S0 (control), S1, S2, and S3 were 0.444, 0.560, 0.694, and 0.741 dB/cm/MHz, respectively, demonstrating a stepwise increase with increasing steatosis severity (P < 0.0001) (S2 Table). Median CAP values for these grades were 196.0, 243.7, 296.3, and 305.3, respectively, also demonstrating a stepwise increase with increasing steatosis severity (P < 0.0001). AUROC curves in diagnosing steatosis grades ≥1, ≥2, and 3 using UGAP or CAP are shown in Table 3. AUROC curve results indicate that UGAP was significantly superior to CAP in diagnosing steatosis grade (P < 0.05).

## Correlation of LSM, AC, and histopathologic evaluation in NAFLD patients, and stratification of NASH or progressive NASH among NAFLD patients

Overall, no significant correlation was found between LSM and AC (Spearman's correlation coefficient r = 0.025, P = 0.725). Fig 2 shows patient distribution according to histopathologic evaluation by LSM and AC.

Fig 3a and 3c shows NASH or progressive NASH patient distribution according to LSM and AC. We divided all NAFLD patients into four groups, according to LSM and AC cutoff values for fibrosis stage ≥F1 and steatosis grade ≥S1, determined by ROC analysis: Group A (n = 14): patients with LSM < 6.430 kPa and AC < 0.493 dB/cm/MHz; Group B (n = 52): patients with LSM < 6.430 kPa and AC ≥ 0.493 dB/cm/MHz; Group C (n = 28): patients with LSM ≥ 6.430 kPa and AC < 0.493 dB/cm/MHz; and Group D (n = 108): patients with LSM ≥ 6.430 kPa and AC ≥ 0.493 dB/cm/MHz. Fig 3b and 3d show percentages of NASH (Fig 3b) and progressive NASH (Fig 3d) patients in Groups A, B, C, and D. In both cases, there was a significant difference in the distribution among groups (P < 0.001).

**Table 1. Baseline characteristics of control participants and patients with NAFLD.**

| Variables | Control | NAFLD | P value |
|---|---|---|---|
| N | 20 | 202 | |
| Sex (male/female) | 10/10 | 99/103 | 0.933 |
| Mean age (years) | 55.3 ± 14.8 | 55.2 ± 12.9 | 0.947 |
| BMI (kg/m$^2$) | 22.2 [20.3–24.4] | 28.8 [25.7–33.3] | < 0.001 |
| Skin-liver capsule distance (mm) | 16.5 [13.8–19.0] | 20.0 [18.0–25.0] | < 0.001 |
| Type II diabetes mellitus (%) | 0 (0.0) | 82 (40.6) | |
| Hypertension (%) | 0 (0.0) | 61 (30.2) | |
| Dyslipidemia (%) | 0 (0.0) | 109 (53.9) | |
| T.Bil (mg/dL) | 0.7 [0.5–0.9] | 0.7 [0.5–0.9] | 0.632 |
| AST (U/L) | 22.0 [18.0–25.0] | 34.0 [21.0–52.0] | < 0.001 |
| ALT (U/L) | 22.5 [17.3–26.0] | 41.0 [21.0–68.5] | < 0.001 |
| Alb (g/dL) | 4.5 [4.3–4.6] | 4.2 [3.9–4.4] | < 0.001 |
| GGT (U/L) | 21.0 [13.5–31.3] | 42.0 [23.0–95.0] | < 0.001 |
| FPG (mg/dL) | 96.0 [91.0–98.8] | 107.5 [91.0–128.8] | 0.003 |
| IRI (μU/mL) | 6.5 [4.8–8.1] | 16.1 [7.3–22.7] | 0.001 |
| HDL-C (mg/dL) | 61.5 [52.5–80] | 52.0 [42.0–61.0] | 0.010 |
| TG (mg/dL) | 82.3 [53.3–122.8] | 107.0 [82.3–147.8] | 0.017 |
| Plt (×10$^4$/mm$^3$) | 19.8 [17.8–22.6] | 19.3 [15.1–24.1] | 0.408 |
| NAFL/NASH | | 73/129 | |
| NAS (1/2/3/4/5/6/7) | | 39/29/37/59/18/10/10 | |
| Fibrosis stage (%) | | | |
| F0 (none) | | 71 (35.1) | |
| F1 (perisinusoidal or periportal) | | 44 (21.8) | |
| F2 (perisinusoidal and portal/periportal))l) | | 32 (15.8) | |
| F3 (bridging fibrosis) | | 28 (13.9) | |
| F4 (cirrhosis) | | 27 (13.4) | |
| Steatosis grade (%) | | | |
| S0 (<5%) | | 0 (0.0) | |
| S1 (5%-33%) | | 142 (72.3) | |
| S2 (34%-66%) | | 43 (21.3) | |
| S3 (>67%) | | 17 (8.4) | |
| Lobular inflammation grade (%) | | | |
| A0 (None) | | 53 (26.2) | |
| A1 (<2 foci per 200× field) | | 82 (40.6) | |
| A2 (2–4 foci per 200× field) | | 63 (31.2) | |
| A3 (>4 foci per 200× field) | | 4 (2.0) | |
| Ballooning grade (%) | | | |
| B0 (none) | | 70 (34.7) | |
| B1 (few balloon cells) | | 100 (49.5) | |
| B2 (many balloon cells) | | 32 (15.8) | |

The values are shown as the mean ± standard deviation or the median [25-75th percentile]. Alb, albumin; ALT, alanine aminotransferase; AST, aspartate aminotransferase; BMI, body mass index; FPG, fasting plasma glucose; GGT, gamma-glutamyl transferase; HDL-C, HDL cholesterol; IRI, immunoreactive insulin; NAFL, non-alcoholic fatty liver; NAFLD, non-alcoholic fatty liver disease; NAS, non-alcoholic fatty liver disease activity score; NASH, non-alcoholic steatohepatitis; Plt, platelet count; T.Bil, total bilirubin; TG, triglyceride.

**Table 2. Diagnostic accuracy of 2D SWE and VCTE for the diagnosis of fibrosis stage.**

| | ≥F1 (95% CI) | | ≥F2 (95% CI) | | ≥F3 (95% CI) | | F4 (95% CI) | |
|---|---|---|---|---|---|---|---|---|
| | 2D SWE | VCTE | 2D SWE | VCTE | 2D SWE | VCTE | 2D SWE | VCTE |
| AUROC | 0.815 (0.760–0.869) | 0.777 (0.689–0.875) | 0.870 (0.823–0.917) | 0.837 (0.789–0.925) | 0.910 (0.872–0.948) | 0.897 (0.805–0.968) | 0.933 (0.892–0.974) | 0.923 (0.863–0.987) |
| Cut-off value | 6.430 | 7.672 | 7.250 | 9.921 | 8.409 | 11.892 | 10.047 | 14.864 |
| Sensitivity | 0.832 (0.758–0.887) | 0.595 (0.510–0.675) | 0.874 (0.785–0.929) | 0.649 (0.564–0.725) | 0.875 (0.760–0.940) | 0.893 (0.781–0.953) | 0.889 (0.709–0.968) | 0.926 (0.753–0.989) |
| Specificity | 0.659 (0.557–0.748) | 0.791 (0.696–0.862) | 0.689 (0.606–0.761) | 0.747 (0.648–0.825) | 0.759 (0.688–0.818) | 0.745 (0.663–0.796) | 0.826 (0.766–0.873) | 0.734 (0.678–0.800) |
| PPV | 0.779 (0.710–0.847) | 0.804 (0.725–0.883) | 0.644 (0.558–0.730) | 0.787 (0.710–0.864) | 0.551 (0.447–0.654) | 0.532 (0.431–0.633) | 0.414 (0.287–0.541) | 0.333 (0.227–0.440) |
| NPV | 0.732 (0.636–0.828) | 0.576 (0.489–0.663) | 0.894 (0.835–0.953) | 0.596 (0.506–0.687) | 0.947 (0.909–0.985) | 0.953 (0.917–0.990) | 0.982 (0.961–1.000) | 0.986 (0.968–1.000) |
| LR+ | 2.443 (1.817–3.284) | 2.852 (1.866–4.358) | 2.808 (2.158–3.654) | 2.567 (1.764–3.736) | 3.631 (2.724–4.841) | 3.369 (2.574–4.409) | 5.098 (3.653–7.114) | 3.611 (2.779–4.692) |
| LR− | 0.255 (0.169–0.383) | 0.511 (0.405–0.646) | 0.184 (0.104–0.323) | 0.470 (0.362–0.610) | 0.165 (0.082–0.331) | 0.146 (0.068–0.312) | 0.135 (0.046–0.392) | 0.112 (0.026–0.379) |

2D SWE, two-dimensional shear wave elastography; AUROC, area under the receiver operating characteristic curve; CI: confidence interval; LR+, positive likelihood ratio; LR−, negative likelihood ratio; NPV, negative predictive value; PPV, positive predictive value; VCTE, vibration controlled transient elastography.

## Predictive factors associated with progressive NASH by univariate and multivariate regression models

We explored predictive factors associated with progressive NASH among baseline parameters. Age, type II diabetes mellitus, aspartate aminotransferase, alanine aminotransferase, gamma-glutamyl transferase, fasting plasma glucose, immunoreactive insulin, high-density lipoprotein cholesterol, triglycerides, LSM, and AC were significant parameters for predicting poor prognosis, as assessed by univariate regression analysis (Table 4). Further, these factors were analyzed using a stepwise multiple regression model, revealing that LSM and AC were independent factors for predicting progressive NASH (Table 4). These two variables were computed into the progressive NASH logistic regression model, establishing the following

**Table 3. Diagnostic accuracy of UGAP and CAP for the diagnosis of steatosis grade.**

| | ≥S1 (95% CI) | | ≥S2 (95% CI) | | S3 (95% CI) | |
|---|---|---|---|---|---|---|
| | UGAP | CAP | UGAP | CAP | UGAP | CAP |
| AUROC | 0.891 * (0.837–0.945) | 0.829 (0.775–0.903) | 0.909 * (0.869–0.949) | 0.832 (0.782–0.902) | 0.924 ** (0.877–0.972) | 0.801 (0.717–0.886) |
| Cut-off value | 0.493 | 241 | 0.654 | 276 | 0.691 | 300 |
| Sensitivity | 0.791 (0.729–0.842) | 0.627 (0.558–0.691) | 0.842 (0.723–0.916) | 0.807 (0.684–0.890) | 0.882 (0.642–0.977) | 0.765 (0.521–0.908) |
| Specificity | 0.900 (0.684–0.982) | 0.950 (0.743–1.000) | 0.866 (0.804–0.910) | 0.774 (0.704–0.832) | 0.833 (0.776–0.878) | 0.721 (0.655–0.778) |
| PPV | 0.988 (0.970–1.000) | 0.992 (0.977–1.000) | 0.686 (0.577–0.794) | 0.554 (0.447–0.661) | 0.306 (0.177–0.435) | 0.186 (0.095–0.277) |
| NPV | 0.300 (0.184–0.416) | 0.202 (0.121–0.283) | 0.940 (0.903–0.978) | 0.920 (0.875–0.965) | 0.988 (0.972–1.000) | 0.974 (0.948–0.999) |
| LR+ | 7.910 (2.120–29.515) | 12.537 (1.850–84.946) | 6.278 (4.188–9.410) | 3.577 (2.622–4.880) | 5.294 (3.721–7.532) | 2.737 (1.941–3.859) |
| LR- | 0.232 (0.171–0.315) | 0.393 (0.320–0.482) | 0.182 (0.100–0.333) | 0.249 (0.146–0.426) | 0.141 (0.038–0.520) | 0.327 (0.138–0.773) |

AUROC, area under receiver the operating curve; CAP, controlled attenuation parameter; CI: Confidence interval; LR+, positive likelihood ratio; LR−, negative likelihood ratio; NPV, negative predictive value; PPV, positive predictive value; UGAP, ultrasound-guided attenuation parameter.

* p < 0.05 (compared with CAP).

** p < 0.01 (compared with CAP).

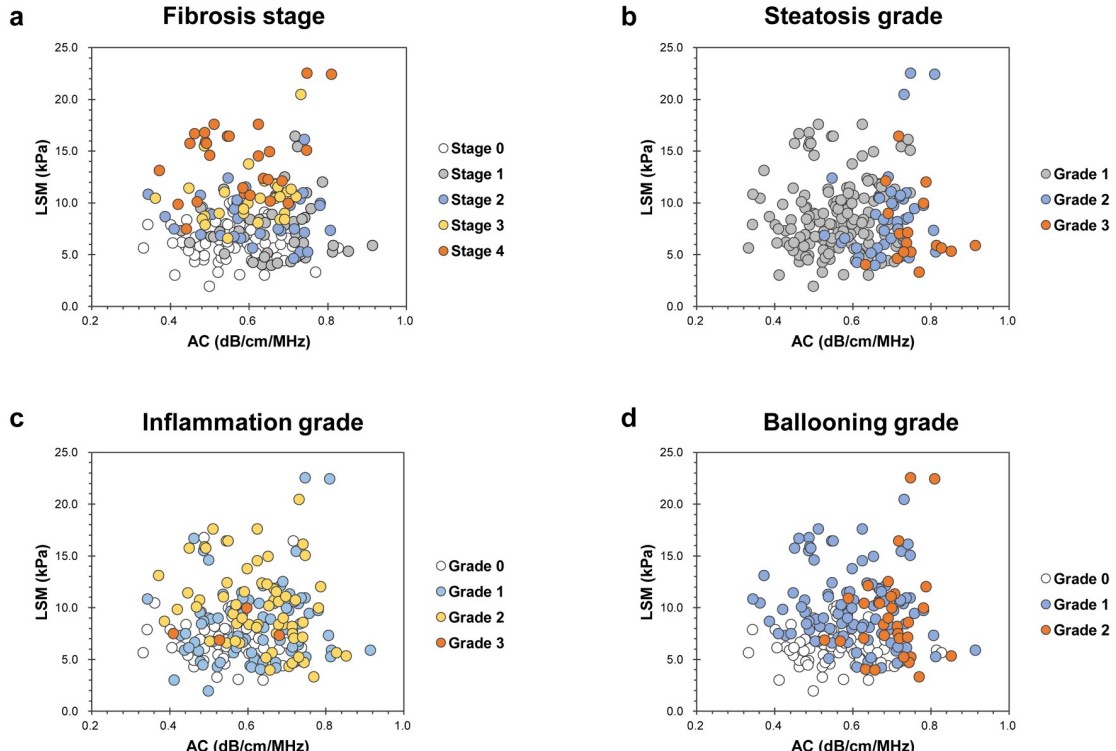

**Fig 2. Patient distribution according to histopathologic evaluation, LSM, and AC.** Fibrosis stage (a), steatosis grade (b), inflammation grade (c), and ballooning grade (d). LSM, liver stiffness measurement; AC, attenuation coefficient.

model established: logit (p) = 0.5414 × LSM (kPa) + 7.791 × AC (dB/cm/MHz)—8.401. The AUC of this prediction model was 0.832 with 80.9% sensitivity, 74.6% specificity, 61.8% PPV, and 88.5% NPV. The Hosmer–Lemeshow test indicated that the prediction model was well-adjusted (P = 0.235). The AUCs were larger for the prediction model than for the FAST score; however, there was no statistically significant difference between the two models (S3 Fig).

## Discussion

Our results confirm that 2D SWE and UGAP are superior for diagnosing fibrosis and steatosis, respectively, in these patients and that their combination is useful for risk stratification of progressive NASH patients.

In our study, 2D SWE diagnostic accuracy was comparable to that of VCTE for liver fibrosis staging in NAFLD patients. However, 2D SWE is not included in current guidelines on NAFLD management because of limited evidence available in NAFLD patients [9,10,34]. We investigated the relationship between degree of steatosis, hepatic inflammatory activity, or ballooned hepatocytes with LSM measured using 2D SWE in NAFLD patients. We found a significant correlation between LSM and grade of hepatic inflammatory activity and ballooned hepatocytes (S1 Table).

Although a steady stepwise increase in AC or CAP values was observed with increasing hepatitis steatosis severity using both UGAP and CAP, UGAP diagnostic accuracy was significantly superior. In addition, UGAP success rate was 100.0%, also superior to that of CAP. Our results suggest that UGAP is advantageous over CAP. First, the UGAP measurement screen is displayed in B-mode, and the operator can easily avoid structures affecting measured values.

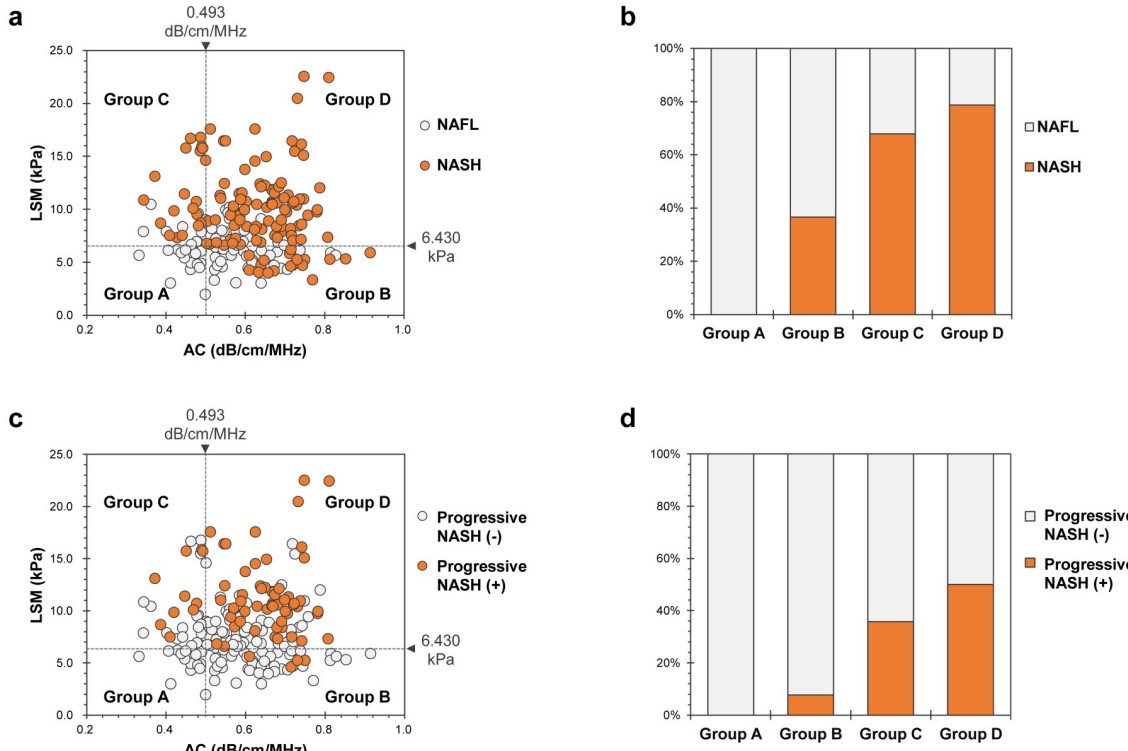

**Fig 3. NASH or progressive NASH patient distribution according to LSM and AC.** a, c: All patients with nonalcoholic fatty liver disease were divided into four groups, according to the cutoff values of LSM and AC for fibrosis stage ≥ F1 and steatosis grade ≥ S1 [NASH (a), progressive NASH (c)]. b, d: Percentage of NASH (b) or progressive NASH (d) patients in each group. Percentages of NASH patients in Groups A, B, C, and D were 0.0% (0/14), 36.5% (19/52), 67.97% (19/28), and 78.7% (85/108), respectively; there was a significant difference in the distribution among groups (P < 0.001) (b). Percentages of progressive NASH patients in Groups A, B, C, and D were 0.0% (0/14), 7.7% (4/52), 35.7% (10/28), and 50.0% (54/108), respectively; there was a significant difference in the distribution among groups (P < 0.001) (d). LSM, liver stiffness measurement; AC, attenuation coefficient.

Second, since the CAP area of interest is fixed, the measurement area may include extrahepatic areas. Finally, CAP uses two probes, whereas UGAP can measure AC with a single probe using traditional ultrasound devices; thus, UGAP is likely advantageous cost-wise. We investigated the relationship between degree of fibrosis, hepatic inflammatory activity, or ballooned hepatocytes and AC. There was a significant correlation between AC and stage of fibrosis, grade of hepatic inflammatory activity, and ballooned hepatocytes (S2 Table).

As effective pharmacological therapy for NASH has not yet been established, we focused on the detection of progressive NASH. In this study, no significant correlation was found between LSM using 2D SWE and AC using UGAP. To clarify the influence of LSM and AC's interaction on progressive NASH risk, we hypothesized that combined evaluation by 2D SWE and UGAP would be useful for assessing the risk of progressive NASH. We subsequently divided the diverse population of NAFLD patients into four groups according to 2D SWE and UGAP results. We believed that this classification may reflect the serial progression from fat deposition within hepatocytes, hepatic inflammation, hepatocyte injury, and fibrogenesis to cirrhosis. Steatosis and fibrosis absence corresponds to low LSM and AC values (Group A). In Group A, risk of NASH or progressive NASH was extremely low, so LB necessity was low; thus, follow-up with diet and lifestyle changes is desirable. A successful outcome for the extraction of Group A may bring substantial benefits to several NAFLD patients. AC value increases with fat deposition within hepatocytes (Group B). In Group B, we believe specialist referral and LB

**Table 4. Predictive factors for progressive NASH, as assessed using univariate and multivariate regression models.**

| Parameter | Univariate Analysis | | Multivariate Analysis | |
|---|---|---|---|---|
| | OR (95% CI) | P value | OR (95% CI) | P value |
| Sex (male) | 0.901 (0.497–1.635) | 0.733 | | |
| Age * | 1.021 (1.004–1.045) | 0.019 | 1.021 (0.990–1.053) | 0.195 |
| BMI | 1.030 (0.923–1.027) | 0.326 | | |
| Skin-liver capsule distance | 1.024 (0.986–1.064) | 0.957 | | |
| Type II diabetes mellitus (+) * | 2.040 (1.135–3.665) | 0.017 | 1.869 (0.988–2.015) | 0.624 |
| Hypertension (+) | 1.457 (0.786–2.701) | 0.231 | | |
| Dyslipidemia (+) | 1.656 (0.866–2.157) | 0.154 | | |
| T.Bil | 1.768 (0.805–3.885) | 0.156 | | |
| AST * | 1.056 (1.032–1.081) | < 0.0001 | 1.011 (0.998–1.024) | 0.088 |
| ALT * | 1.010 (1.015–1.015) | < 0.0001 | | |
| Alb | 0.733 (0.374–1.012) | 0.364 | | |
| GGT * | 1.007 (1.002–0.677) | 0.010 | 1.002 (0.996–1.009) | 0.490 |
| FPG * | 1.021 (1.008–1.035) | 0.002 | 1.012 (0.996–1.030) | 0.151 |
| IRI * | 1.116 (1.065–1.170) | < 0.0001 | 1.031 (0.989–1.075) | 0.146 |
| HDL-C * | 0.962 (0.943–0.982) | < 0.0001 | 0.963 (0.934–1.015) | 0.098 |
| TG * | 1.006 (1.001–1.012) | 0.021 | 0.543 (0.991–1.005) | 0.543 |
| Plt | 0.998 (0.993–1.003) | 0.344 | | |
| LSM (2D SWE) ** | 1.626 (1.377–1.920) | < 0.0001 | 1.555 (1.279–1.891) | < 0.0001 |
| AC **, *** | 1.734 (1.297–2.391) | < 0.0001 | 1.615 (1.037–2.517) | 0.024 |

2D SWE, two-dimensional shear wave elastography; 95% CI, 95% confidence interval; AC, attenuation coefficient; Alb, albumin; ALT, alanine aminotransferase; AST, aspartate aminotransferase; BMI, body mass index; FPG, fasting plasma glucose; GGT, gamma-glutamyl transferase; HDL-C, high-density lipoprotein cholesterol; IRI, immunoreactive insulin; LSM, liver stiffness measurement; NASH, non-alcoholic steatohepatitis; OR, odds ratio; Plt, platelet count; T.Bil, total bilirubin; TG, triglyceride.

\* Significant factor, as assessed with univariate analysis.

\*\* Significant factor, according to both univariate and multivariate analyses.

\*\*\* For an increment of 10 dB/cm/MHz.

necessity are high because NASH or progressive NASH is highly possible. Moreover, the need for diet and lifestyle changes may be high in Group B patients. In steatohepatitis, ballooning degeneration indicative of hepatocyte necrosis, inflammatory cell infiltration, and pericellular fibrosis increases LSM value (Group C or D). Inevitably, Group C and D patients have higher risks of NASH or progressive NASH, so specialist referral necessity is high. In Group D, the percentage of patients with NAS ≥ 4 was 62.9% (68/108), and there were many cases of high activity. Thus, LB necessity for NASH diagnosis may be low, but necessity of a prompt start of therapeutic intervention may be high (S1 Table).

In this study, we proposed a prediction model for progressive NASH risk in NAFLD patients. Progressive NASH patients had significantly higher LSM and AC values, which were determined as independent factors for predicting progressive NASH. In our analysis, the difference between progressive NASH and other was most significant at the cutoff point of 0.30. Thus, we suggest that a prediction model value more than 0.30 on the day of NAFLD diagnosis indicates progressive disease. These patients will require effective pharmacological therapy in the future. Our new prediction model utilizing ultrasound parameters is a simple and useful method for predicting progressive NASH prognosis, showing equivalent diagnostic ability to the FAST score. This prediction model may reduce the number of patients having unnecessary LB.

On the other hand, the model for end-stage liver disease (MELD) was originally developed for the assessment of short-term mortality in patients with cirrhosis, and its clinical uses have been since extended to include prioritization of liver transplantation [35,36]. Matthews et al. recently showed that the MELD score is associated with presence of cardiovascular disease in a large cross-sectional NAFLD cohort [37]. Therefore, as a sub-analysis, we analyzed the correlation between our proposed prediction model for progressive NASH and MELD score, finding a positive correlation (Spearman's correlation coefficient r = 0.334, P < 0.0001). Our prediction model utilizing ultrasound parameters may be effective in predicting the prognosis of NAFLD and requires further accumulation of cases and detailed analyses.

Our prospective study has several strengths. Firstly, 2D SWE, UGAP, VCTE, CAP, and LB were performed on the same day to avoid bias. Secondly, a simple prediction model was used for progressive NASH utilizing ultrasound parameters. Finally, the four quadrants generated by 2D SWE and UGAP easily allow progressive NASH risk assessment in each group. These four quadrants may be more acceptable to clinicians than the logistic regression equation as they are simple and intuitive.

However, this study has several limitations. First, larger-scale prospective clinical studies in Japan and possibly in different Asian countries are needed to confirm our findings. Second, the presence of an ultrasound device is required for 2D SWE and UGAP assessments. Use of these applications is limited by high cost of equipment and need for well-trained operators. Third, no cases of burned-out NASH were enrolled in this study. The characteristic features of NASH disappear in advanced cirrhosis (i.e., burned-out NASH). We did not include these cases because they never had LB. We speculate that burned-out NASH would be assigned to Group C. Thus, a study design including such cases should be considered in future studies.

In conclusion, our findings suggest that 2D SWE and UGAP are superior in diagnosing fibrosis and steatosis, respectively, in NAFLD patients, and that their combined assessment is useful for progressive NASH risk stratification and may be convenient in evaluating specialist referral and liver biopsy necessity.

## Supporting information

**S1 Fig. Two-dimensional shear wave elastography performed on a fatty liver.** The colored box (center) represents the elastogram, and the circle (yellow) represents the region of interest where the elastic modulus (liver stiffness measurement) of the liver is acquired. The blue color indicates soft liver tissue, as semi-quantitatively presented by the color scale to the left. (TIF)

**S2 Fig. Ultrasound-guided attenuation parameter (UGAP).** The attenuation coefficient (AC) was calculated based on the reference phantom method reported by Yao et al. [24]. This method utilizes an ultrasound phantom with known attenuation (AC: 0.44 dB/cm/MHz) and backscatter coefficients to compensate for the characteristics of transmission. The echo signal from the liver $S_0(f, x)$ (target) and the echo signal from the phantom, $S_p(f, x)$ (reference) are described as

$$-\frac{1}{2f} \log_{10} \frac{s_0(f, x)}{s_p(f, x)} + \alpha_p x = \alpha_0 x \qquad (1)$$

where $f$ is the frequency used, $x$ is the length of the depth direction of the region of interest (ROI), and $\alpha_0$ and $\alpha_p$ are the ACs of the tissue and phantom, respectively. The ultrasound system was calibrated using a specific acquisition set up (4.0 MHz of the fundamental B-mode) before the study, and the same acquisition setup was used for collecting the data for each patient. The results of the onetime calibration were used to calculate the AC. We acquired

B-mode image data of the liver parenchyma (Segment 5), the same area subjected to two-dimensional shear wave elastography, vibration controlled transient elastography, controlled attenuation parameter, and liver biopsy (a). Radiofrequency-based ultrasound echo signals were analyzed by a dedicated prototype software program in MATLAB (MathWorks, Inc., Natick, MS, USA). One of three ultrasound engineers (T.O., S.N., and N.K.) opened each image and set the ROI on the liver parenchymal area, avoiding vessels, and at least 20 mm from the liver surface. The engineers were blinded to all patient data. The average of 10 consecutive scanning rasters was processed, followed by smoothing using a low-pass filter (b). The AC was calculated based on the signal's decay slope A between 55 and 120 mm in depth using the least-squares method (c):

$$\frac{1}{2f} log_{10} \frac{s_0(f,x)}{s_p(f,x)} + \alpha_p x \approx Ax + B \tag{2}$$

(TIF)

**S3 Fig. Receiver operating characteristic (ROC) analysis for the prediction model of progressive non-alcoholic steatohepatitis (NASH).** The area under the ROC curve (AUROC) for the prediction of progressive NASH was 0.832 for the prediction model, and 0.824 for the FAST score. There was no statistically significant difference between the two models. Abbreviations: AC: attenuation coefficient; AUROC, area under the receiver operating characteristic curve; FAST: FibroScan-aspartate aminotransferase; LSM, liver stiffness measurement; LR+, positive likelihood ratio; LR-, negative likelihood ratio NASH, non-alcoholic steatohepatitis; NPV, negative predictive value; PPV, positive predictive value.
(TIF)

**S1 Table. 2D SWE and VCTE measurement results by each histopathologic evaluation.**
(DOCX)

**S2 Table. UGAP and CAP measurement results by each histopathologic evaluation.**
(DOCX)

**S3 Table. Risk stratification of progressive NASH and the necessity of liver biopsy.**
(DOCX)

## Acknowledgments

The authors thank Ms. Yuriko Mikami and Ms. Koko Motodate for their excellent technical assistance.

## Author Contributions

**Conceptualization:** Hidekatsu Kuroda.

**Data curation:** Hidekatsu Kuroda, Yudai Fujiwara, Tamami Abe, Tomoaki Nagasawa.

**Formal analysis:** Hidekatsu Kuroda, Yudai Fujiwara, Tamami Abe, Tomoaki Nagasawa.

**Funding acquisition:** Hidekatsu Kuroda.

**Investigation:** Hidekatsu Kuroda, Yudai Fujiwara, Tamami Abe, Tomoaki Nagasawa.

**Methodology:** Hidekatsu Kuroda, Yudai Fujiwara, Tamami Abe, Tomoaki Nagasawa.

**Project administration:** Hidekatsu Kuroda, Tamami Abe, Yasuhiro Takikawa.

**Resources:** Hidekatsu Kuroda, Yudai Fujiwara, Tamami Abe, Tomoaki Nagasawa.

**Software:** Takuma Oguri, Sachiyo Noguchi, Naohisa Kamiyama.

**Supervision:** Yasuhiro Takikawa.

**Validation:** Hidekatsu Kuroda, Yudai Fujiwara, Tamami Abe, Tomoaki Nagasawa, Yasuhiro Takikawa.

**Visualization:** Hidekatsu Kuroda, Yasuhiro Takikawa.

**Writing – original draft:** Hidekatsu Kuroda.

**Writing – review & editing:** Hidekatsu Kuroda, Yasuhiro Takikawa.

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
