## [Decision Letter · Decision Letter 0]

9 Feb 2021

PONE-D-21-01341

Two-dimensional shear wave elastography and ultrasound-guided attenuation parameter for progressive non-alcoholic steatohepatitis

PLOS ONE

Dear Dr. Hidekatsu Kuroda,

Thank you for submitting your manuscript to PLOS ONE. After careful consideration, we feel that it has merit but does not fully meet PLOS ONE’s publication criteria as it currently stands. Therefore, we invite you to submit a revised version of the manuscript that addresses the points raised during the review process. The study has merit and we encourage a resubmission.

Please submit your revised manuscript within 60 days. If you will need more time than this to complete your revisions, please reply to this message or contact the journal office at plosone@plos.org. Please include the following items when submitting your revised manuscript:

We look forward to receiving your revised manuscript.

Kind regards,

Gianfranco D. Alpini

Academic Editor

PLOS ONE

Journal Requirements:

"NO authors have competing interests"

We note that one or more of the authors are employed by a commercial company: GE Healthcare.

(2) Please also provide an updated Competing Interests Statement declaring this commercial affiliation along with any other relevant declarations relating to employment, consultancy, patents, products in development, or marketed products, etc.  

3. Please upload a copy of Supporting Information S1 which you refer to in your text on pages 30 and 40.

Reviewers' comments:

Reviewer's Responses to Questions

**Comments to the Author**

1. Is the manuscript technically sound, and do the data support the conclusions?

Reviewer #1: Yes

Reviewer #2: Yes

2. Has the statistical analysis been performed appropriately and rigorously? 

Reviewer #1: Yes

Reviewer #2: Yes

3. Have the authors made all data underlying the findings in their manuscript fully available?

Reviewer #1: Yes

Reviewer #2: Yes

4. Is the manuscript presented in an intelligible fashion and written in standard English?

Reviewer #1: Yes

Reviewer #2: Yes

5. Review Comments to the Author

Reviewer #1: In the current study, Kuroda et al. aimed to evaluate the effectiveness of two-dimensional shear wave elastography (2D-SWE) and ultrasound-guided attenuation parameter (UGAP) for progressive non-alcoholic steatohepatitis (NASH). They recruited 233 NAFLD patients but later excluded 18 patients from the study due to other known etiologies. They first compared the diagnostic accuracy of 2D-SWE in fibrosis stages with the vibration-controlled attenuation parameter (VCTE). It was found that the 2D-SWE may be slightly better than VCTE. They then compared the diagnostic accuracy of UGAP in steatosis grade with the controlled attenuation parameter (CAP). It was noted that the UGAP is significantly better in steatosis grade diagnosis than CAP. Besides, the liver stiffness measurement (LSM) calculated by using 2D-SWE and VCTE is significantly correlated with the liver fibrosis stage. They also found that the attenuation coefficient (AC) calculated by using UGAP and the CAP are correlated with steatosis grades. Furthermore, the LCM and AC could be both used as predictive factors for progressive NASH. This is an interesting clinical study with significance to the field of NAFLD. However, several issues identified in this manuscript should be further clarified.

1. On page 17, it was mentioned that median LSMs obtained by 2D-SWE and VCTE in NAFLDs are correlated with liver fibrosis stages. However, the materials and methods section only mentioned that LSM was performed using 2D-SWE. How was LSM calculation using VCTE performed?

2. The data of LSM scores and fibrosis stages should be listed as a table.

3. The data of AC and steatosis grade as well as CAP and steatosis grade should be listed as a table.

4. Figures 2, 3a, and 3c are somehow confusing. You suggest that LSM and AC are correlated with disease progression. However, the figures indicate that LSM and AC have no correlation which may not necessarily support your findings.

5. In tables 2 and 3, the alignment of each row should be adjusted. It is confusing based on the current format.

6. Did you measure 2D-SWE and UGAP in the control group? Did you incorporate the data with your NAFLD group?

Reviewer #2: In this study, the authors aimed to investigate the usefulness of combining two-dimensional shear wave elastography and the ultrasound-guided attenuation parameter for assessing the risk of progressive non-alcoholic steatohepatitis. The authors demonstrated that two-dimensional shear wave elastography and vibration-controlled transient elastography had equivalent diagnostic performance for fibrosis. Combined assessment by two-dimensional shear wave elastography and the ultrasound-guided attenuation parameter is useful for risk stratification of progressive non-alcoholic steatohepatitis. Overall, data provided here by the authors could have interesting implications. Specific points need to be considered are listed below:

1. It’s necessary to briefly introduce the technique and examination procedure of UGAP in the Methods section.

2. It would be interesting to evaluate the relationship between Model for End-Stage Liver Disease (MELD) score with the two-dimensional shear wave elastography and ultrasound-guided attenuation parameter.

6. PLOS authors have the option to publish the peer review history of their article (what does this mean?). If published, this will include your full peer review and any attached files.

Reviewer #1: No

Reviewer #2: No

---

## [Author Response · Author response to Decision Letter 0]

4 Mar 2021

March 1, 2021 

Emily Chenette

Deputy Editor-in-Chief

PLOS ONE

Dear Editor: 

We wish to resubmit our manuscript titled “Two-dimensional shear wave elastography and ultrasound-guided attenuation parameter for progressive non-alcoholic steatohepatitis” (PONE-D-21-01341) for publication in PLOS ONE. 

We sincerely thank the editors and reviewers for their critical review, constructive comments, and valuable suggestions, which have been incorporated in our revised manuscript and enabled us to provide, what we believe, is a significantly improved account of our research. Our point-by-point responses to the comments of the reviewers are appended below. The revisions in the manuscript are denoted by underlined text, as specified. 

In accordance with your instructions, we would like to update our Funding Statement as follows: This work was supported by JSPS KAKENHI [grant number JP 19K08400]. The funder provided support in the form of salaries for authors [H.K.]. However, it did not have any additional role in the study design, data collection, and analysis, decision to publish, or preparation of the manuscript. The specific roles of these authors are articulated in the ‘author contributions’ section.

In addition, we would like to update our Competing Interests Statement as follows: The authors have read the journal’s policy, and the authors of this manuscript have the following competing interests: T.O., S.N. and N.K. are paid employees of GE Healthcare Japan. This does not alter our adherence to PLOS ONE policies on sharing data and materials. The other authors declare that they have no competing interests. There are no products in development or marketing products to declare.

We thank you for your consideration and look forward to hearing from you.

Sincerely yours,

Hidekatsu Kuroda, M.D.

Division of Hepatology, Department of Internal Medicine, Iwate Medical University, Nishitokuta2-1-1, Yahaba-cho, Shiwa-gun, Iwate, 028-3694, Japan

E-mail: hikuro@iwate-med.ac.jp

Telephone: +81-19-651-5111

Fax: +81-19-907-7166

Our responses to the comments from reviewers 1 and 2 are as follows:

RESPONSES TO REVIEWER 1

We thank the reviewer for the thoughtful and constructive review of our manuscript. We have addressed all of the reviewer's concerns in detail. Our responses to the reviewer’s comments are as follows:

1) On page 17, it was mentioned that median LSMs obtained by 2D-SWE and VCTE in NAFLDs are correlated with liver fibrosis stages. However, the materials and methods section only mentioned that LSM was performed using 2D-SWE. How was LSM calculation using VCTE performed?

Response:

We thank the reviewer for their comments on this point. In accordance with the reviewer's advice, we have revised the text in the Materials and methods section (Materials and methods: Pages 9–10, lines 155–172).

2) The data of LSM scores and fibrosis stages should be listed as a table.

Response:

We fully agree with your helpful suggestion. Accordingly, we have presented these data in a table (S1 Table, 2D SWE and VCTE measurement results by each histopathologic evaluation).

3) The data of AC and steatosis grade as well as CAP and steatosis grade should be listed as a table.

Response:

We fully agree with your helpful suggestion. Accordingly, we have presented these data in a table (S2 Table. UGAP and CAP measurement results by each histopathologic evaluation).

4) Figures 2, 3a, and 3c are somehow confusing. You suggest that LSM and AC are correlated with disease progression. However, the figures indicate that LSM and AC have no correlation which may not necessarily support your findings.

Response:

We thank the reviewer for this comment. As you pointed, we found no significant correlation between LSM using 2D SWE and AC using UGAP. However, we wish to present in Figures 2 and 3 the distribution of ultrasonic parameters according to the pathological findings of NAFLD / NASH and the usefulness of classification using two parameters. Initially, to clarify the influence of LSM and AC's interaction on progressive NASH risk, we hypothesized that combined evaluation by 2D SWE and UGAP would be useful in determining the risk of progressive NASH. We divided the diverse population of NAFLD patients into four groups, according to 2D SWE and UGAP results, and determined the difference in the distribution of NASH or progressive NASH among the four groups. Furthermore, in the logistic regression analysis, LSM and AC were independently associated with progressive NASH. We have revised the relevant text in the manuscript (Discussion: Page 30, lines 369–377).

5) In tables 2 and 3, the alignment of each row should be adjusted. It is confusing based on the current format.

Response:

We fully agree with the reviewer and have formatted the tables accordingly (Table 2 and 3).

6) Did you measure 2D-SWE and UGAP in the control group? Did you incorporate the data with your NAFLD group?

Response:

We thank the reviewer for this insightful comment. We performed 2D-SWE and UGAP analyses of the control group and described the results in the text and tables (Materials and methods: Page 7, lines 119–121; S1 Table; S2 Table.). The data of the control group were not incorporated into those of the NAFLD group.

RESPONSES TO REVIEWER 2

We thank the reviewer for their thoughtful and constructive examination of our manuscript. We have addressed all of the reviewer's concerns in detail. Our responses to the reviewer’s comments are as follows:

1) It’s necessary to briefly introduce the technique and examination procedure of UGAP in the Methods section.

Response:

We thank the reviewer for raising this point. According to the reviewer's advice, we have revised the text in the Materials and methods section (Materials and methods: Pages 8–9, lines 140–153).

2) It would be interesting to evaluate the relationship between Model for End-Stage Liver Disease (MELD) score with the two-dimensional shear wave elastography and ultrasound-guided attenuation parameter.

Response:

We completely agree with the reviewer’s valuable suggestion. We analyzed the correlation between our proposed prediction model for progressive NASH and MELD score, finding a positive correlation (Spearman's correlation coefficient r = 0.334, P < 0.0001). We have revised the relevant text in the manuscript (Discussion: Page 32, lines 401–410).

---

## [Decision Letter · Decision Letter 1]

19 Mar 2021

Two-dimensional shear wave elastography and ultrasound-guided attenuation parameter for progressive non-alcoholic steatohepatitis

PONE-D-21-01341R1

Dear Dr. Hidekatsu Kuroda,

We’re pleased to inform you that your manuscript has been judged scientifically suitable for publication and will be formally accepted for publication once it meets all outstanding technical requirements.

Kind regards,

Gianfranco D. Alpini

Academic Editor

PLOS ONE

Additional Editor Comments (optional):

Reviewers' comments:

Reviewer's Responses to Questions

**Comments to the Author**

1. If the authors have adequately addressed your comments raised in a previous round of review and you feel that this manuscript is now acceptable for publication, you may indicate that here to bypass the “Comments to the Author” section, enter your conflict of interest statement in the “Confidential to Editor” section, and submit your "Accept" recommendation.

Reviewer #1: All comments have been addressed

Reviewer #2: All comments have been addressed

2. Is the manuscript technically sound, and do the data support the conclusions?

Reviewer #1: Yes

Reviewer #2: (No Response)

3. Has the statistical analysis been performed appropriately and rigorously? 

Reviewer #1: Yes

Reviewer #2: (No Response)

4. Have the authors made all data underlying the findings in their manuscript fully available?

Reviewer #1: Yes

Reviewer #2: (No Response)

5. Is the manuscript presented in an intelligible fashion and written in standard English?

Reviewer #1: Yes

Reviewer #2: (No Response)

6. Review Comments to the Author

Reviewer #1: Thanks for the clarification. All concerns have been addressed by the authors. It is acceptable to be published in the PLOS ONE journal.

Reviewer #2: (No Response)

7. PLOS authors have the option to publish the peer review history of their article (what does this mean?). If published, this will include your full peer review and any attached files.

Reviewer #1: No

Reviewer #2: No

---

## [Editor Report · Acceptance letter]

24 Mar 2021

PONE-D-21-01341R1 

Two-dimensional shear wave elastography and ultrasound-guided attenuation parameter for progressive non-alcoholic steatohepatitis 

Dear Dr. Kuroda:

I'm pleased to inform you that your manuscript has been deemed suitable for publication in PLOS ONE. Congratulations! Your manuscript is now with our production department. 

Kind regards, 

on behalf of

Dr. Gianfranco D. Alpini 

Academic Editor

PLOS ONE